# Information Value: Measuring Utterance Predictability as Distance from Plausible Alternatives

**Mario Giulianelli**[◁][*] **Sarenne Wallbridge**[◇][*] **Raquel Fernández**[◁]

[◁]Institute for Logic, Language and Computation, University of Amsterdam
[◇]Centre for Speech Technology Research, University of Edinburgh
m.giulianelli@uva.nl  s1301730@ed.ac.uk  raquel.fernandez@uva.nl

## Abstract

We present *information value*, a measure which quantifies the predictability of an utterance relative to a set of plausible alternatives. We introduce a method to obtain interpretable estimates of information value using neural text generators, and exploit their psychometric predictive power to investigate the dimensions of predictability that drive human comprehension behaviour. Information value is a stronger predictor of utterance acceptability in written and spoken dialogue than aggregates of token-level surprisal and it is complementary to surprisal for predicting eye-tracked reading times.[1]

## 1 Introduction

When viewed as information transmission, successful language production can be seen as an act of reducing the uncertainty over future states that a comprehender may be anticipating. Saying a word, for example, may cut the space of possibilities in half, while uttering a whole sentence may restrict the comprehender's expectations to a far smaller space. Measuring the amount of information carried by a linguistic signal is fundamental to the computational modelling of human language processing. Such quantifications are used in psycholinguistic and neurobiological models of language processing (Levy, 2008; Willems et al., 2016; Futrell and Levy, 2017; Armeni et al., 2017), to study the processing mechanisms of neural language models (Futrell et al., 2019; Davis and van Schijndel, 2020; Sinclair et al., 2022), and as a learning and evaluation criterion for language modelling (under the guise of 'cross-entropy loss' or 'perplexity'). The amount of information carried by a linguistic signal is intrinsically related to its predictability (Hale, 2001; Genzel and Charniak, 2002; Jaeger and Levy, 2007). This connection is summarised in the definition of the *surprisal*, or

*information content*, of a unit $u$ (Shannon, 1948), perhaps the most widely used measure of information: $I(u) = -\log_2 p(u)$. Predictable units carry low amounts of information—i.e., low surprisal—as they are already expected to occur given the context in which they are produced. Conversely, unexpected units carry higher surprisal.

Proper estimation of the surprisal of an utterance is intractable, as it would require computing probabilities over a high-dimensional, structured, and ultimately unbounded event space. It is thus common to resort to chaining token-level surprisal estimates, nowadays typically obtained from neural language models (Meister et al., 2021; Giulianelli and Fernández, 2021; Wallbridge et al., 2022). However, token-level autoregressive approximations of utterance probability have a few problematic properties. A well-known issue is that different realisations of the same concept or communicative intent compete for probability mass (Holtzman et al., 2021), which implies that the surprisal of semantically equivalent realisations is overestimated. Moreover, token-level surprisal estimates conflate different dimensions of predictability. As evidenced by recent studies (Arehalli et al., 2022; Kuhn et al., 2023), this makes it difficult to appreciate whether the information carried by an utterance is a result, for example, of the unexpectedness of its lexical material, syntactic arrangements, semantic content, or speech act type.

We propose an intuitive characterisation of the information carried by utterances, *information value*, which computes predictability over the space of full utterances to account for potential communicative equivalence, and explicitly models multiple dimensions of predictability (e.g., lexical, syntactic, and semantic), thereby offering greater interpretability of predictability estimates. Given a linguistic context, the *information value* of an utterance is a function of its distance from the set of contextually expected alternatives. The intuition is

---

[*]Shared first authorship.

[1]https://github.com/dmg-illc/information-value

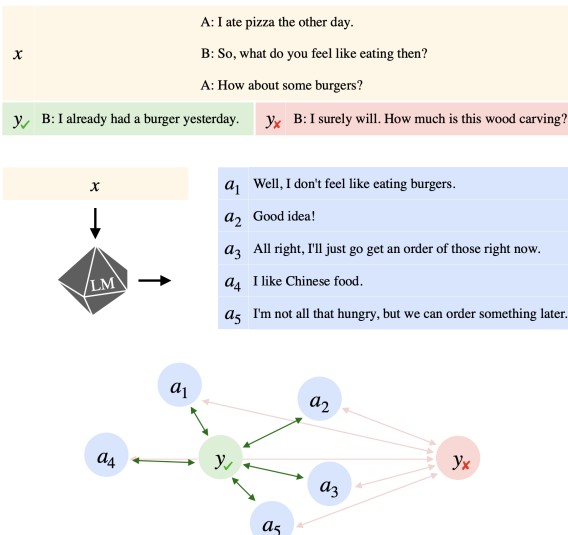

Figure 1: The information value $I(Y = y | X = x)$ of a target utterance $y$ is lower—and its predictability higher—when $y$ is closer to the set of plausible alternatives $A_x = (a_1, a_2, \ldots)$. Here, alternatives are generated by an LM conditioned on a context $x$.

that if an utterance differs largely from alternative productions, it is an unexpected contribution to discourse with high information value (see Figure 1). We obtain empirical estimates of information value by sampling alternatives from neural text generators and measuring their distance from a target utterance using interpretable distance metrics. Information value estimates are evaluated in terms of their ability to predict and explain human reading times and acceptability judgements in dialogue and text.

We find information value to have stronger psychometric predictive power than aggregates of token-level surprisal for acceptability judgements in spoken and written dialogue, and to be complementary to surprisal aggregates as a predictor of reading times. Furthermore, we use our interpretable measure of predictability to gather insights into the processing mechanisms underlying human comprehension behaviour. Our analysis reveals, for example, that utterance acceptability in dialogue is largely determined by semantic expectations while reading times are more affected by lexical and syntactic predictions.

Information value is a new way to measure predictability. As such, next to surprisal, it is a powerful tool for the analysis of comprehension behaviour (Meister et al., 2021; Shain et al., 2022; Wallbridge et al., 2022, 2023), for the computational modelling of language production strategies (Doyle and Frank, 2015; Xu and Reitter, 2018;

Verma et al., 2023) and for the design of processing and decision-making mechanisms that reproduce them in natural language generation systems (Wei et al., 2021; Giulianelli, 2022; Meister et al., 2023).

## 2 Background

**Surprisal theory.** Expectation-based theories of language processing define the effort required to process a linguistic unit as a function of its predictability. Surprisal theory, perhaps the most prominent example, posits a direct relationship between effort and predictability, quantified as surprisal (Hale, 2001). The theory is supported by broad empirical evidence across domains and languages (Pimentel et al., 2021; de Varda and Marelli, 2022), and serves as a foundation for quantitative principles of language production and comprehension such as Entropy Rate Constancy (ERC; Genzel and Charniak, 2002) and Uniform Information Density (UID; Levy and Jaeger, 2007).

**The psychometric predictive power of surprisal.** Without direct access to the 'true'[2] conditional probabilities of linguistic units, psycholinguists have relied on statistical models of language to estimate surprisal (Hale, 2001; McDonald and Shillcock, 2003). More recently, large-scale language models have emerged as powerful estimators of token-level surprisal, reflected by their ability to predict different aspects of human language comprehension behaviour (their *psychometric predictive power*). Psychometric variables include self-paced and eye-tracked reading times (Keller, 2004; Goodkind and Bicknell, 2018; Wilcox et al., 2020; Meister et al., 2021; Shain et al., 2022; Oh and Schuler, 2023), acceptability judgements (Lawrence et al., 2000; Heilman et al., 2014; Lau et al., 2015, 2017; Warstadt et al., 2019; Wallbridge et al., 2022), and brain response data (Frank et al., 2015; Schrimpf et al., 2021).

To obtain estimates of utterance surprisal, different aggregates of token-level surprisal have been proposed, motivated by psycholinguistic theories like ERC and UID. However, their behaviour is far less understood (e.g., Wallbridge et al., 2022). For example, divergences between how model characteristics affect predictive power for different comprehension tasks (e.g., Meister et al., 2021) raise questions about whether token-level

---

[2]In this context, *true* refers to the—if at all existing— unattainable conditional probabilities of linguistic units that a human may experience during language comprehension.

aggregates appropriately capture expectations over utterances in human language processing.

**Alternatives in semantics and pragmatics.** Our proposed notion of information value takes inspiration from the concept of alternatives in semantics and pragmatics (Horn, 1972; Grice, 1975; Stalnaker, 1978; Gazdar, 1979; Rooth, 1996; Levinson, 2000). Reasoning about alternatives has been argued to be at the basis of the use of questions (Hamblin, 1976; Groenendijk and Stokhof, 1984; Ciardelli et al., 2018), focus (Rooth, 1992; Wagner et al., 2005; Beaver and Clark, 2009), and implicatures (Carston, 1998; Degen and Tanenhaus, 2015, 2016; Zhang et al., 2023). Recently, alternative sets generated with the aid of language models have been used to provide empirical evidence that pragmatic inferences of scalar implicature depend on listeners' context-driven uncertainty over alternatives (Hu et al., 2022, 2023). Hu et al. (2022) generate sets of plausible words in context, within scalar constructions, then embed and cluster the resulting sentences to simulate conceptual alternatives. Reasoning over word- and concept-level alternatives is operationalised through surprisal and entropy. To our knowledge, ours is the first study to use language models for the generation of full utterance-level alternatives.

## 3   Alternative-Based Information Value

Given a context $x$, a speaker may produce a number of plausible utterances. We refer to these as $A_x$, the *alternative set*. We define the *information value* of an utterance $y$ in a context $x$ as the real random variable which captures the distribution of distances between $y$ and the set of alternative productions $A_x$, measured with a distance metric $d$:

$$I(Y\!=\!y|X\!=\!x) \coloneqq d(y, A_x) \qquad (1)$$

This distribution characterises the predictability of $y$ in its context. Large distances indicate that $y$ differs substantially from expected utterances, and thus that $y$ is a surprising next utterance.

### 3.1   Computing Information Value

In Equation 1, we define information value as a statistical measure of the unpredictability, or unexpectedness of an utterance. In practice, computing the information value of an utterance requires (1) a method for obtaining alternative sets $A_x$, (2) a metric with which to measure the distance of an utterance from its alternatives, and (3) a means with which to summarise distributions of pairwise distances. We discuss these three elements in turn in the following paragraphs.

**Generating alternative sets.** Since the 'true' alternative sets entertained by a human comprehender are not attainable, we propose generating them algorithmically, via neural text generators. Being able to guarantee the plausibility, or humanlikeness of the generations is crucial. Our approach builds on recent work (Giulianelli et al., 2023) finding the predictive distribution of neural text generators to be well aligned to human variability, as measured with the same distance metrics used in this paper (see next paragraph): while not all generations are guaranteed to be of high quality, their low-dimensional statistical properties (e.g., $n$-gram, POS, and speech act distribution) match those of human productions. This should allow us to obtain faithful distance distributions $d(y, A_x)$ and thus accurate estimates of information value.

**Measuring distance from alternatives.** We quantify the distance of a target utterance from an alternative production using three interpretable distance metrics, as defined by Giulianelli et al. (2023). **Lexical:** Fraction of distinct $n$-grams in two utterances, with $n \in [1, 2, 3]$ (i.e., the number of distinct $n$-gram occurrences divided by the total number of $n$-grams in both utterances). **Syntactic:** Fraction of distinct part-of-speech (POS) $n$-grams in two utterances. **Semantic:** Cosine and euclidean distance between the sentence embeddings of two utterances (Reimers and Gurevych, 2019). These distance metrics characterise alternative sets at varying levels of abstraction (Katzir, 2007; Fox and Katzir, 2011; Buccola et al., 2022), enabling an exploration into the representational form of expectations over alternatives in human language processing.

**Summarising distance distributions.** Information value is a random variable that describes a distribution over distances between an utterance $y$ and the set of plausible alternatives (Equation 1). To summarise this distribution, we explore *mean* as the expected distance (under a uniform distribution over alternatives) or as the distance from a prototypical alternative, and *min* as the distance of $y$ from the closest alternative production, implicating that proximity to a single alternative is sufficient to determine predictability.

## 4 Experimental Setup

### 4.1 Language Models

We generate alternative sets using neural autoregressive language models (LMs). For the dialogue corpus, we use GPT-2 (Radford et al., 2019), DialoGPT (Zhang et al., 2020), and GPT-Neo (Black et al., 2021). For the text corpora, we use GPT-2, GPT-Neo, and OPT (Zhang et al., 2022). The text models are pre-trained, while dialogue models are fine-tuned on the respective datasets. Further details on fine-tuning and perplexity scores are in Appendix A. The resulting dataset, which contains 1.3M generations, is publicly available.[3]

**Generating alternatives.** To generate an alternative set $A_x$, we sample from $p_{LM}(Y|X = x)$. We experiment with four popular sampling algorithms to ensure that the quality of our information value estimates is not dependent on a particular algorithm—or, if it is, that we are not overlooking it. We select (1) *unbiased* (ancestral or forward) sampling (Bishop, 2006; Koller and Friedman, 2009), (2) *temperature sampling* ($\alpha \in [0.75, 1.25]$), (3) *nucleus sampling* (Holtzman et al., 2019) ($p \in [0.8, 0.85, 0.9, 0.95]$), and (4) *locally typical sampling* (Meister et al., 2023) ($\tau \in [0.2, 0.3, 0.85, 0.95]$), for a total of 11 sampling strategies. We post-process alternatives to ensure that each contains only a single utterance.[4]

### 4.2 Psychometric Data

Using five corpora, we study two main types of psychometric variables that rely on different underlying processing mechanisms (Gibson and Thomas, 1999; Hofmeister et al., 2014): acceptability judgements and reading times.

#### 4.2.1 Acceptability judgements

Stimuli for acceptability judgements typically consist of isolated sentences that are manipulated automatically or by hand to assess a *grammatical* notion of acceptability (Lau et al., 2017; Warstadt et al., 2019). The effect of context on acceptability is still relatively underexplored, yet contextualised judgements arguably capture a more natural, intuitive notion of acceptability. In this study, we use some of the few datasets of in-context acceptability judgements which examine grammaticality as well as semantic and pragmatic plausibility.

SWITCHBOARD and DAILYDIALOG. Participants were presented with a short sequence of dialogue turns followed by a potential upcoming turn, and asked to rate its plausibility in context on a scale from 1 to 5. Judgements were collected by Wallbridge et al. (2022) for (transcribed) spoken dialogue and written dialogue from the Switchboard Telephone Corpus (Godfrey et al., 1992) and DailyDialog (Li et al., 2017), respectively. For each corpus, 100 items are annotated by 3-6 participants. Annotation items consist of 10 dialogue contexts, each followed by the true next turn and by 9 turns randomly sampled from the respective corpus.[5]

CLASP. Participants were presented with sentences from the English Wikipedia in and out of their document context and asked to judge acceptability using a 4-point scale (Bernardy et al., 2018). The original sentences are round-trip translated into 4 languages to obtain varying degrees of acceptability; the context is not modified. This dataset contains 500 stimuli, annotated by 20 participants.[6]

#### 4.2.2 Reading times

Previous literature regarding the predictive power of language models for reading behaviour has focused on the relationship between per-word surprisal and reading times (Keller, 2004; Wilcox et al., 2020; Shain et al., 2022; Oh and Schuler, 2023). We define utterance-level reading time as the total time spent reading the constituent words of the utterance. This approach has been taken by previous studies of utterance-level surprisal (Meister et al., 2021; Amenta et al., 2022).

PROVO. This corpus consists of 136 sentences (55 paragraphs) of English text from a variety of genres. Eye movement data was collected from 84 native American English speakers (Luke and Christianson, 2018). We use the summation of word-level reading times (IA-DWELL-TIME, the total duration of all fixations on the target word) of constituent words to obtain utterance-level measures.

BROWN. This corpus consists of self-paced moving-window reading times for 450 sentences (12 passages) from the Brown corpus of American

---

[3]*AltGen*: https://doi.org/10.5281/zenodo.10006413.

[4]We use spaCy's sentence segmentation algorithm (Honnibal et al., 2020) for the text corpora and split dialogue utterances based on the position of the turn separator.

[5]Acceptability ratings available at https://data.cstr.ed.ac.uk/sarenne/INTERSPEECH2022/.

[6]We only use judgements collected in context, available at https://github.com/GU-CLASP/BLL2018.

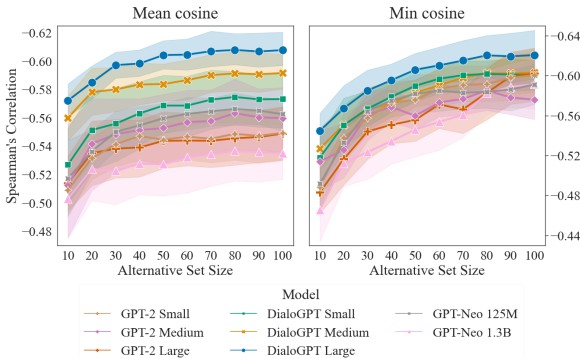

| | Information value | Surprisal |
|---|---|---|
| *Acceptability* $(x \propto y^{-1})$ | | |
| SWITCHBOARD | -0.702 *(semantic)* | -0.506 *(superlinear, $k=4$)* |
| DAILYDIALOG | -0.584 *(semantic)* | -0.457 *(superlinear, $k=2.5$)* |
| CLASP | -0.234 *(syntactic)* | -0.559 *(mean)* |
| *Reading times* $(x \propto y)$ | | |
| PROVO | 0.421 *(syntactic)* | 0.495 *(variance)* |
| BROWN | 0.223 *(lexical)* | 0.220 *(mean)* |

Table 1: Correlations of the most predictive variants (in parentheses) of surprisal and information value across model, sampling algorithm, and alternative set size with psychometric data: mean acceptability judgements and length-normalised reading times.

Figure 2: Spearman correlation between semantic information value and mean acceptability judgements in SWITCHBOARD. Confidence intervals display variability over 11 sampling strategies.

English. Reading times were collected from 35 native English speakers (Smith and Levy, 2013).

# 5 Psychometric Predictive Power

We begin by evaluating our empirical estimates of information value in terms of their psychometric predictive power: can they predict comprehension behaviour recorded as human acceptability judgements and reading times? We test the robustness of this predictive power to the alternative set generation process and compare it to previously proposed utterance-level surprisal aggregates including mean, variance, and a range of summation strategies; see Appendix B for full definitions.

For each corpus in Section 4.2, we measure the correlation between information value and the respective psychometric variable, which is the average in-context acceptability judgement for DAILYDIALOG, SWITCHBOARD, and CLASP, and the total utterance reading time normalised by utterance length for PROVO and BROWN.[7] Alternative sets are generated using the language models and sampling strategies described in Section 4.1. Lexical, syntactic, and semantic distances are computed in terms of the distance metrics presented in Section 3.1, for alternative sets of varying size ($[10, 20, ..., 100]$). The distributions of similarities in Equation 1 are summarised using *mean* and *min*, thus yielding scalar estimates of information value.

## 5.1 Predictive Power

For every corpus, we obtain a moderate to strong Spearman correlation between information value and the psychometric target variable. For example, estimates of semantic information value correlate with acceptability judgements at strengths approximately between $-0.4$ and $-0.7$ for SWITCHBOARD and between $-0.3$ and $-0.6$ for DAILYDIALOG across models and sampling strategies from Section 4.1 (see Figure 2 for SWITCHBOARD; Appendix C for all datasets). Estimates obtained with the best information value estimators for each corpus, shown in Table 1, yield substantially higher correlations with acceptability in dialogue than the best token-level aggregates of utterance surprisal, both as computed in our experiments and as reported in prior work (Wallbridge et al., 2022, 2023). Reading times, on the other hand, which are aggregates of word-level psychometric data points, should naturally be easier to capture with word-level measures of predictability. Nevertheless, our best information value estimates correlate with reading times only slightly less strongly or comparably to surprisal; and additionally, they give us indications about the dimensions of unexpectedness (in this case, lexical and syntactic) that mostly affect reading behaviour.

Overall, beyond building trust in our information value estimators, this evaluation demonstrates the benefit of their interpretability. The predictive power for lexical, syntactic, and semantic distances varies widely between corpora. Semantic distances are much more predictive for dialogue datasets than lexical or syntactic distances, while the inverse is true for the reading times datasets. We explore differences between the underlying perceptual processes employed for these two comprehension tasks further in Section 6.

---

[7]We normalise by utterance length as it is an obvious correlate of total reading time and would have confounding effects on this analysis. In Section 6, we confirm our findings using mixed effect models that include utterance length as a predictor and total unnormalised reading time as a response variable.

## 5.2 Robustness to Estimator Parameters

We now study the extent to which our estimates are affected by variation in three important parameters of alternative set generation: the alternative set size ($[10, 20, ..., 100]$), the language model, and the sampling strategy. We find a slight positive, asymptotic relationship between predictive power, reflected by correlations between information value and psychometric data, and alternative set size for semantic information value in the dialogue corpora—information value estimates become more predictive as alternative set size increases (see, e.g., Figure 2). Set size does not significantly affect correlations for the reading times corpora. Moreover, while we do observe differences between models, and larger models tend to obtain higher correlations with psychometric variables, these results are not consistent across corpora and distance metrics (Figures 4 and 5, Appendix C). In light of recent findings regarding the *inverse* relationship between language model size and the predictive power of surprisal (Shain et al., 2022; Oh and Schuler, 2023), we consider it an encouraging result that the predictive power of information value does not decrease with the number of model parameters.[8] We do not observe a significant impact of decoding strategy on predictive power, regardless of alternative set size, as indicated by the confidence intervals in Figures 2, 4 and 5.

In sum, estimates of information value do not display much sensitivity to alternative set generation parameters.[9] Therefore, for each corpus, we select the estimator (a combination of model, sampling algorithm, and alternative set size) that yields the best Spearman correlation with the psychometric data (Table 5 in Appendix E). We use these estimators throughout the rest of the paper.

## 6 In-Depth Analysis of Psychometric Data

Using information value, we now study which dimensions of predictability effectively explain psychometric data. This allows us to qualitatively analyse the processes humans employ while reading and assessing acceptability. We also examine the effect of contextualisation on comprehension behaviour by defining two additional measures derived from information value (Section 6.1) and

---

[8]It remains to be seen whether this trend extends to larger language models, for which we lack computational resources.

[9]We obtain similar evidence of robustness to parameter settings using an intrinsic evaluation, reported in Appendix D.

---

using them as explanatory variables in linear mixed effect models to predict per-subject psychometric data. For the dialogue corpora and CLASP, our mixed effect models predict in-context acceptability judgements. For the reading times corpora, our models predict the total time spent by a subject reading a sentence, as recorded in self-paced reading and eye-tracking studies. This is the sum, over a sentence, of word-level reading times (more details in Appendix F). We include random intercepts for *(context, target)* pairs in all models.

**Analysis Procedure.** For every corpus, we first test models that include a single predictor beyond the baseline: i.e., information value measured with each distance metric and either *mean* and *min* as summary statistics (see Section 3.1). Based on the fit of these single-predictor models, we select the best lexical, syntactic, and semantic distance metrics (with the corresponding summary statistics) to instantiate three-predictor models for each of the derived measures of information value.

Following Wilcox et al. (2020), we evaluate each model relative to a baseline model which includes only control variables. Control variables are selected building on previous work (Meister et al., 2021): solely the intercept term for acceptability judgements and the number of fixated words for reading times (more details in Appendix F). As an indicator of explanatory power, we report $\Delta$LogLik, the difference in log-likelihood between a model and the baseline: a positive $\Delta$LogLik value indicates that the psychometric variable is more probable under the comparison model. We also report fixed effect coefficients and their statistical significance. The full results are shown in Table 6 (Appendix F), according to which the best metrics for each linguistic level are selected and used throughout the rest of the paper.

## 6.1 Derived Measures of Information Value

Inspired by information-theoretic concepts used in previous work to study the predictability of utterances (e.g., Genzel and Charniak, 2002; Giulianelli and Fernández, 2021; Wallbridge et al., 2022), we define two additional derived measures of information value and assess their explanatory power.

*Out-of-context information value* is the distance between an utterance $y$ and the set of alternative productions $A_\epsilon$ expected given the empty context $\epsilon$:

$$I(Y = y) := I(Y = y | X = \epsilon) \qquad (2)$$

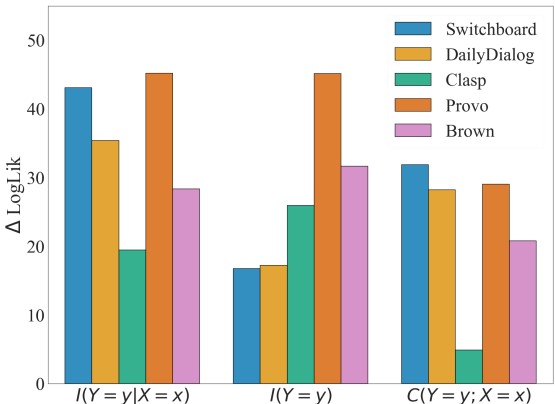

Figure 3: Explanatory power of information value and its derived measures (out-of-context information value and context informativeness; defined in Section 6.1).

It reflects the plausibility of $y$ regardless of its context of occurrence. An analogous notion is decontextualised surprisal.

*Context informativeness* is the reduction in information value for $y$ contributed by context $x$:

$$C(Y=y; X=x) := I(Y=y) - I(Y=y|X=x) \tag{3}$$

This quantifies the extent to which a context restricts the space of plausible productions such that $y$ becomes more predictable. An analogous notion is the pointwise mutual information.

## 6.2 Acceptability

We generally expect an inverse relationship between information value and in-context acceptability judgements: information value is lower when a target utterance is closer to the set of alternatives a comprehender may expect in a given context (see Figure 1). Furthermore, we expect grammaticality and semantic plausibility—two factors known to affect acceptability (Sorace and Keller, 2005; Lau et al., 2017)—to play different roles in dialogue and text. For the dialogue corpora, we expect semantic-level variables to have high explanatory power, as they can identify utterances with incoherent content such as implausible underlying dialogue acts (Searle, 1969, 1975; Austin, 1975). Lexical and syntactic information value may be more explanatory of acceptability in CLASP, where stimuli are generated via round-trip translation and thus may contain disfluent or ungrammatical sentences (Somers, 2005).

**SWITCHBOARD and DAILYDIALOG.** For both dialogue corpora, semantic information content

is by far the most predictive variable (Table 6, Appendix F), especially when *min* is used as a summary statistic. Responses to the same dialogue context can exhibit great variability and being close to a single expected alternative—in terms of semantic content and dialogue act type—appears to be sufficient for an utterance to be considered acceptable. Our analysis of derived measures (Figure 3) further indicates that acceptability is mostly determined by the in-context predictability of an utterance. The high explanatory power of context informativeness (almost twice that of out-of-context information value) suggests that contextual cues override inherent isolated plausibility.

**CLASP.** Syntactic information value is the best explanatory variable for acceptability judgements in CLASP (Table 6, Appendix F). This suggests that comprehenders entertain expectations over syntactic structures (here, represented as POS sequences)—a result which could complement findings on the processing of lexicalised constructions in reading (e.g., Tremblay et al., 2011) and eye-tracking studies (e.g., Underwood et al., 2004). In contrast to the dialogue corpora, estimates of in-context information value are less predictive than their out-of-context counterparts (Figure 3), which may be due to the previously discussed artificial nature of the CLASP negative samples. In sum, our results indicate that the acceptability judgements in the CLASP corpus, even if collected in context, are mostly determined by the presence of startling surface forms rather than by semantic expectations.

## 6.3 Reading Times

When reading, humans continually update their expectations about how the discourse might evolve (Hale, 2001; Levy, 2008; Yan and Jaeger, 2020). This is reflected, for example, in the faster processing of more expected words and syntactic structures (Demberg and Keller, 2008; Smith and Levy, 2013). High predictive power for lexical and syntactic information value would support these findings. However, comprehenders also reason about semantic alternatives, e.g., to compute scalar inferences (Van Tiel et al., 2014; Hu et al., 2023). Our interpretable measures of information value help clarify the contribution of different types of expectations.

**PROVO and BROWN.** Syntactic information value is a strong predictor of eye-tracked reading times in PROVO, while lexical information value (in particular, based on trigram distances) is the

|  | SWITCHBOARD | DAILYDIALOG | PROVO |
|---|---|---|---|
| **Surprisal** | 6.63 | 5.08 | 59.04 |
| **Information value** | | | |
| *Lexical* | 8.32 | 10.88 | 12.17 |
| *Syntactic* | 2.49 | 6.71 | 21.80 |
| *Semantic* | 34.20 | 30.41 | 6.86 |
| *All* | 43.11 | 35.42 | 45.19 |
| **Joint** | | | |
| *+ Lexical* | 14.08 | 10.23 | 72.60 |
| *+ Syntactic* | 9.77 | 8.05 | 75.70 |
| *+ Semantic* | 34.37 | 26.98 | 68.61 |
| *+ All* | 44.11 | 30.55 | 93.08 |

Table 2: $\Delta$LogLik for surprisal, information value, and joint mixed effect models. We report the best information value metrics (as per Section 6) and surprisal aggregates for each dataset: maximum for SWITCH-BOARD, and super-linear for DAILYDIALOG ($k = 1.5$) and PROVO ($k = 0.5$).

only significant explanatory variable for the self-paced reading times in BROWN (Table 6), and only weakly so. Expectations over full semantic alternatives have a limited effect on reading times in both corpora, suggesting anticipatory processing mechanisms at play during reading operate at lower linguistic levels. For both corpora, out-of-context estimates are at least as predictive as in-context estimates and higher than context informativeness (Figure 3), indicating that context modifications only moderately dampen the negative effects of unusual syntactic arrangements and lexicalised constructions on reading speed.

## 7 Relation to Utterance Surprisal

We have shown alternative-based information value to be a powerful predictor for contextualised acceptability judgements and reading times. In fact, information value is substantially more predictive of acceptability than utterance surprisal (Section 5). We conclude with a focused comparison between these measures, considering whether they are complementary and why they might diverge.

### 7.1 Complementarity

Differences in predictive power between information value and surprisal (see Table 1) may reflect variations between the dimensions of predictability captured by the two measures. To investigate this possibility, we use both measures jointly for psychometric predictions. We focus on the dialogue corpora and PROVO, where we observed the highest explanatory power for information value (Section 6). For each corpus, we fit linear mixed effect models with control variables, using

the most predictive surprisal and information value estimators (one per linguistic level) in isolation and jointly as fixed effects. Table 2 summarises the results of this analysis.

In isolation, information value is a better predictor for the dialogue corpora. Including lexical, syntactic, and semantic information value on top of the best surprisal predictor (*Joint*) improves model log-likelihood substantially. Separately including each linguistic level reveals that semantic distance is largely responsible for improved fit, suggesting that surprisal fails to capture expectations over high-level linguistic properties of utterances such as speech act type, which are crucial for modelling contextualised acceptability in dialogue. This is true regardless of the aggregation function used.

For PROVO, surprisal is the best explanatory variable. However, including the best information value predictors further improves model fit by $58\%$, demonstrating the complementarity of the two measures in predicting reading times (Table 2). Separately adding information value predictors shows the strongest boost comes from syntactic factors, which are known to have higher weight in human anticipatory processing than in language models' (Arehalli et al., 2022).

Overall, combining predictive information value with surprisal yields better models for all tested corpora, indicating that these measures capture distinct and complementary dimensions of predictability.

### 7.2 Effects of Discourse Context

While language comprehension is known to be a function of context (e.g., Kleinschmidt and Jaeger, 2015; Chen et al., 2023), little attention has been given to its impact on surprisal estimates. We examine whether the dissimilar predictability estimates of information value and surprisal stem from differences in their sensitivity to context, comparing how they behave under congruent, incongruent, and empty context conditions. In each condition, alternative sets and token-level surprisal are computed in the true context (*congruent*), a context randomly sampled from the respective corpus (*incongruent*), or with no conditioning (*empty*) as used to compute out-of-context information value[10]. We quantify effects on the best information value and surprisal predictors as $\Delta$LogLik, using single-predictor models described in Section 6.

---

[10]To ensure that all stimuli in this analysis are contextualised, first sentences in PROVO paragraphs were excluded.

| Dataset | Summ. | Level | Metric | Context Condition | | |
|---|---|---|---|---|---|---|
| | | | | Congruent | Empty | Incongruent |
| Switchboard | Mean | Lexical | Trigram | **8.32** | 5.55 | 7.18 |
| | Mean | Syntactic | POS Trigram | 2.49 | **3.00** | 2.65 |
| | Min | Semantic | cosine | **34.20** | 7.64 | 10.94 |
| | Surprisal (in context, max) | | | **6.63** | 2.56 | 3.12 |
| DailyDialog | Min | Lexical | Bigram | **10.88** | 3.16 | 1.42 |
| | Mean | Syntactic | POS Unigram | 6.71 | **6.89** | 6.16 |
| | Min | Semantic | Cosine | **30.41** | 1.43 | 2.90 |
| | Surprisal (in context, superlinear $k=1.5$) | | | **5.08** | 0.99 | 2.35 |
| Provo | Mean | Lexical | Trigram | **12.97** | 12.94 | 11.86 |
| | Mean | Syntactic | POS Trigram | **25.86** | 15.20 | 12.94 |
| | Mean | Semantic | Euclidean | 8.53 | **10.88** | 8.33 |
| | Surprisal (in context, superlinear $k=0.5$) | | | 35.75 | 37.88 | **39.00** |

Table 3: $\Delta$LogLik of single-predictor models for information value and surprisal across context conditions.

Table 3 displays results for Switchboard, DailyDialog, and Provo. Congruent context produces a substantial effect on the predictive power of semantic information value for both dialogue datasets; for DailyDialog, we see a 20-fold increase over the empty context condition. Surprisal shows a similar trend, though far less pronounced. Syntactic information value is the least affected by context modulations. Though surprisal is a powerful predictor for reading times in Provo, the incongruent and empty context conditions are *more* predictive than the true context. Perhaps most concerning is the fact that estimates in incongruent contexts are the most predictive. In contrast, the most predictive information value (syntactic) is significantly more predictive for congruent contexts. Interestingly, information value in the control conditions is not uninformative, likely reflecting the inherent plausibility of utterances.

Both information value and utterance surprisal display sensitivity to context, however, the effects on surprisal are less predictable and perhaps even undesirable for certain psychometric variables.

# 8 Discussion and Conclusion

Humans constantly monitor and anticipate the trajectory of communication. Their expectations over the upcoming communicative signal are influenced by factors spanning from the immediate linguistic context to their interpretation of the speaker's goals. These expectations, in turn, determine aspects of language comprehension such as processing cost, as well as strategies of language production. We present *information value*, a measure which quantifies the predictability of an utterance relative to a set of plausible alternatives; and we introduce a method to obtain information value estimates via neural text generators. In contrast to utterance predictability estimates obtained by aggregating token-level surprisal, information value captures variability above the word level by explicitly accounting for more abstract communicative units like speech acts (Searle, 1969, 1975; Austin, 1975). We validate our measure by assessing its psychometric predictive power, its robustness to parameters involved in the generation of alternative sets, and its sensitivity to discourse context.

Using interpretable measures centred around information value, we investigate the underlying dimensions of uncertainty in human acceptability judgements and reading behaviour. We find that acceptability judgements factor in base rates of utterance acceptability (likely associated with grammaticality) but are predominantly driven by semantic expectations. In contrast, reading time is more influenced by the inherent plausibility of lexical items and part-of-speech sequences. We further compare information value to aggregates of token-level surprisal, finding differences in the dimensions of predictability captured by each measure and their sensitivity to context. Information value is a stronger predictor of acceptability in written and spoken dialogue and is complementary to surprisal for predicting eye-tracked reading times.

Information value is defined in terms of plausible continuations of the current linguistic context, taking inspiration from the tradition of alternatives in semantics and pragmatics (Horn, 1972; Grice, 1975; Stalnaker, 1978). Although the ideal set of alternatives would be derived directly from humans, neural text generators have demonstrated their potential to act as useful proxies, particularly when multiple generations are considered. Variability among their productions has been shown to align with human variability (Giulianelli et al., 2023), and decision rules that operate over sets of alternative utterances, rather than next tokens, have been shown to improve generation quality (e.g., Eikema and Aziz, 2022; Guerreiro et al., 2023). We release our full set of 1.3M generated alternatives, obtained with a variety of models and sampling algorithms, to facilitate research in this direction.

Our information value framework allows considerable flexibility in defining alternative set generation procedures, distance metrics, and summary statistics. We hope it will enable further investigation into the mechanisms involved in human language processing, and that it will serve as a basis for cognitively inspired learning rules and inference algorithms in computational models of language.

## Limitations

Our framework for the estimation of utterance information value allows great flexibility. Modellers can experiment with a variety of alternative set generation procedures, distance metrics, and summary statistics. While our selection of distance metrics characterises the relation of an utterance to its alternative sets at multiple interpretable linguistic levels, there is a large space of metrics that we have not tested in this paper. Syntactic distances, for example, can be computed using metrics that capture structural differences between utterances in a more fine-grained manner (e.g., tree edit distance or difference in syntactic tree depth); semantic distances can be computed with a more taxonomical approach (e.g., Fellbaum, 2010) or using NLI models to capture semantic equivalence (Kuhn et al., 2023); and distances between dialogue act types can be detected using dialogue act classifiers (Stasaski and Hearst, 2023). We chose metrics based on prior work validating them as probes for the extraction of uncertainty estimates from neural text generators (Giulianelli et al., 2023), but we hope future work will explore this space more exhaustively. Similarly, though the current work has been constrained to English data, our framework can be directly applied to other languages. We hope to see work in this direction.

Moreover, due to computational constraints, we selected a single information value estimator per corpus for our analyses in Sections 6 and 7. Although we assessed the sensitivity of information value to parameters of alternative set generation extensively in Appendices C and D, the effect of estimator parameters on the explanatory power of information value predictors can be assessed more widely in future work.

A further aspect of our method for the estimation of information value that we have not highlighted in the paper is its computational cost. Because it involves drawing multiple full utterance samples from language models, our method is clearly less efficient than traditional surprisal estimation, which requires only a single forward pass. While we have observed that the psychometric predictive power of information value reaches satisfactory levels even with relatively low numbers of alternatives and small language model architectures (see, e.g., Figure 2), designing more efficient methods for the estimation of information value is an important direction for future research.

## Ethics Statement

In the Limitations section, we have mentioned that an important direction for future work is designing more computationally efficient methods for the estimation of information value. This is crucial to the application of this method to larger datasets, which may be prohibitively expensive in some research communities and in any case, perhaps unnecessarily, environmentally unfriendly.

The limited size of the corpora of psychometric data used in this paper has further ethical implications, as the corpora have not been collected to be representative of a wide and diverse range of comprehenders. We hope to see efforts in this direction.

## Acknowledgements

We thank the ILLC's Dialogue Modelling Group for helpful comments and discussions. MG and RF are supported by the European Research Council (ERC) under the European Union's Horizon 2020 research and innovation programme (grant agreement No. 819455).

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

# A   Language Models

For the dialogue corpora, we use GPT-2 (Radford et al., 2019), DialoGPT (Zhang et al., 2020), and GPT-Neo (Black et al., 2021). For the text corpora, we use GPT-2 (Radford et al., 2019), GPT-Neo (Black et al., 2021), and OPT (Zhang et al., 2022). The text models are pre-trained while the dialogue models are fine-tuned for 5 epochs with early stopping on the respective datasets, using " " as a turn separator. Preliminary experiments on the pre-trained models show that   is the turn separator that yields lowest perplexity on the dialogue datasets. For text models, using no separator is the option that yields the lowest perplexity. When generating out of context, we set $x$ to be either the dialogue turn separator " " or a white space for the text models.

**LM validation: perplexity.**   Table 4 reports the perplexity of these models on the SWITCHBOARD and DAILYDIALOG test sets, as well as on the WikiText test set (the CLASP dataset and the reading

| | DailyDialog | Switchboard | WikiText |
|---|---|---|---|
| GPT-2 Small (124M) | 7.34 | 11.86 | 25.62 |
| GPT-2 Medium (355M) | 6.03 | 10.50 | 19.69 |
| GPT-2 Large (774M) | 5.26 | 10.09 | 17.39 |
| GPT-Neo 125M | 7.39 | 12.54 | 25.37 |
| GPT-Neo 1.3B | 4.94 | 10.11 | 14.01 |
| DialoGPT Small | 7.94 | 12.50 | - |
| DialoGPT Medium | 6.53 | 10.96 | - |
| DialoGPT Large | 6.23 | 11.00 | - |
| OPT 125M | 17.80 | 22.68 | 46.85 |
| OPT 350M | 14.88 | 21.46 | 40.39 |
| OPT 1.3B | 12.58 | 20.30 | 27.45 |

Table 4: Language model perplexity results. The models tested on the dialogue datasets are finetuned for 5 epochs with early stopping; the models tested on WikiText are pre-trained.

time datasets are too small to allow for robust evaluation, but their style is sufficiently similar enough to that of WikiText). Perplexity scores are the lowest for the dialogue datasets. This is to be expected as the dialogue models are fine-tuned. The perplexity of the pre-trained models on WikiText is in line with state-of-the-art results; OPT obtains higher perplexity than GPT-2 and GPT-Neo, but still in an appropriate range.

## B   Utterance-Level Surprisal

Given an utterance $\mathbf{y}$ as a sequence of tokens in a context $\mathbf{x}$, token-level surprisal can be defined as $s(y_t) = -\log p(y_t|\mathbf{y}_{<t}, \mathbf{x})$. Multiple works have proposed quantifying utterance-level surprisal as functions of token-level surprisal (Genzel and Charniak, 2002; Keller, 2004; Xu and Reitter, 2018; Meister et al., 2021; Giulianelli et al., 2021; Wallbridge et al., 2022). We compare the predictive power of information value to a number of these utterance-level surprisal aggregates.

*Mean surprisal* and *total surprisal* account for all token-level surprisal estimates with and without normalising by utterance length:

$$S_{mean}(\mathbf{y}|\mathbf{x}) = \frac{1}{N} \sum_{n=1}^{N} [s(y_n)] \qquad (4)$$

$$S_{total}(\mathbf{y}|\mathbf{x}) = \sum_{n=1}^{N} [s(y_n)] \qquad (5)$$

*Superlinear surprisal* posits a superlinear effect of token-level estimates:

$$S_{superlinear_k}(\mathbf{y}|\mathbf{x}) = \sum_{n=1}^{N} [s(y_n)]^k \qquad (6)$$

We experiment with $k \in [0.5, 0.75, \ldots, 5]$.

*Maximum surprisal* captures the idea that a highly surprising element drives the overall surprisal of an utterance:

$$S_{max}(\mathbf{y}|\mathbf{x}) = \max[s(y_n)] \qquad (7)$$

Surprisal variance across an utterance has been defined in a number of ways; we consider *surprisal variance* as the regression to the utterance-level mean and *surprisal difference* as the variability between contiguous token-level estimates:

$$S_{variance}(\mathbf{y}|\mathbf{x}) = \frac{1}{N-1} \sum_{n=2}^{N} [s(y_n) - S_{mean}(\mathbf{y})]^2 \qquad (8)$$

$$S_{difference}(\mathbf{y}|\mathbf{x}) = \sum_{n=2}^{N} |s(y_n) - s(y_{n-1})| \qquad (9)$$

## C   Psychometric Predictive Power and Sensitivity of Information Value Estimates

We study the extent to which our estimates of information value are affected by variation in three main factors: the alternative set size ($[10, 20, ..., 100]$), the language model, and the sampling strategy. Figures 4 and 5 show Spearman correlation between information value and psychometric data, averaged over subjects. These results complement Sections 5.1 and 5.2 in the main paper.

## D   Intrinsic Robustness Analysis

In Section 5.1, we evaluate the robustness of information value to parameters involved in the alternative set generation in terms of its psychometric predictive power. We additionally assess their intrinsic robustness by measuring the correlation between information values assigned to target utterances by estimators with different parameter settings.

The parameters which we consider are alternative set size ($[10, 20, ..., 100]$), the generative model, and the decoding strategy. Models and decoding strategies are detailed in Section 4.1. For each of the corpora described in Section 4.2, we compute the information value for the target utterances based on alternative sets generated under different parameter settings. Robustness is quantified through the distribution of the pairwise Spearman

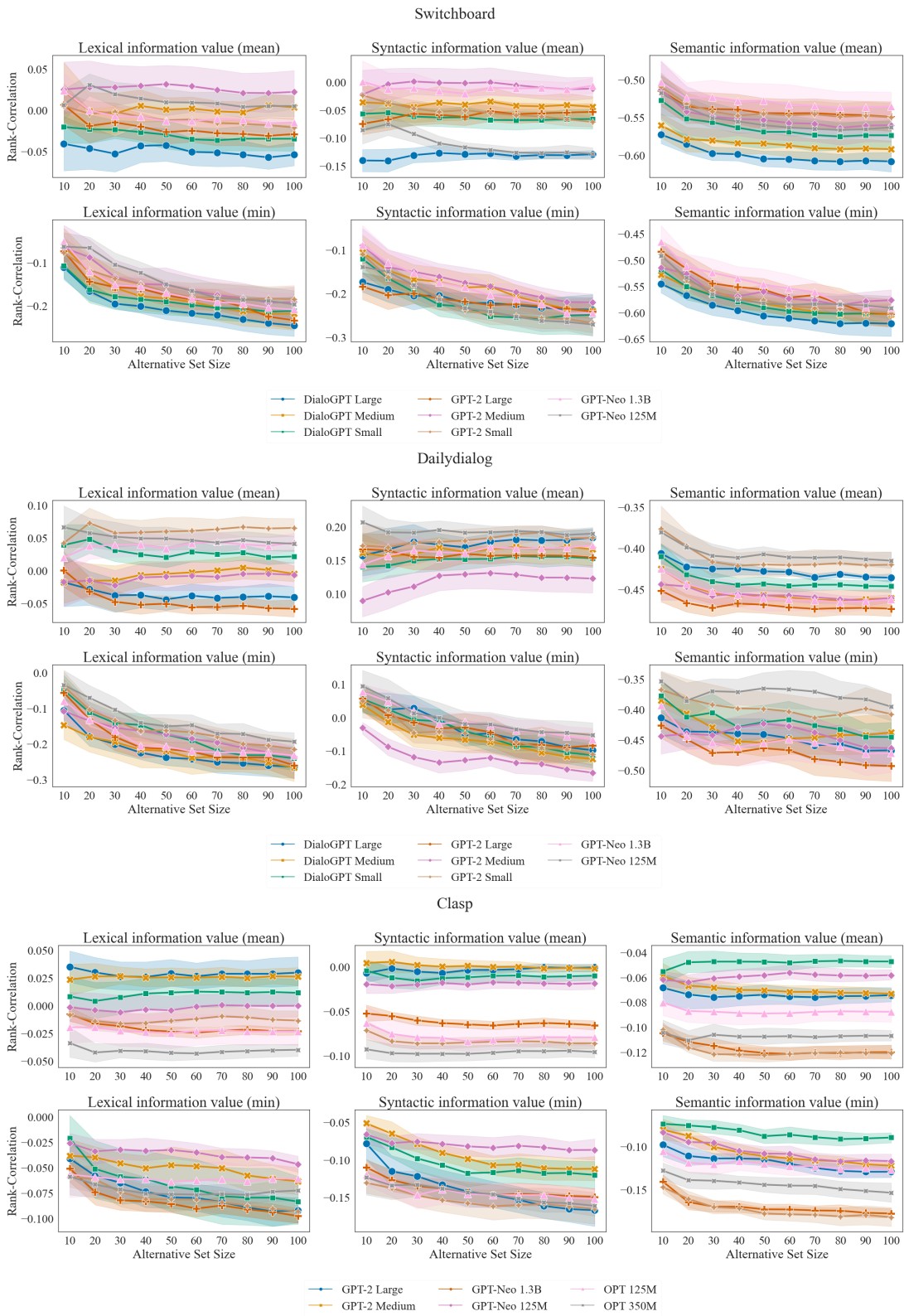

Figure 4: Spearman correlation between information value and average acceptability judgements.

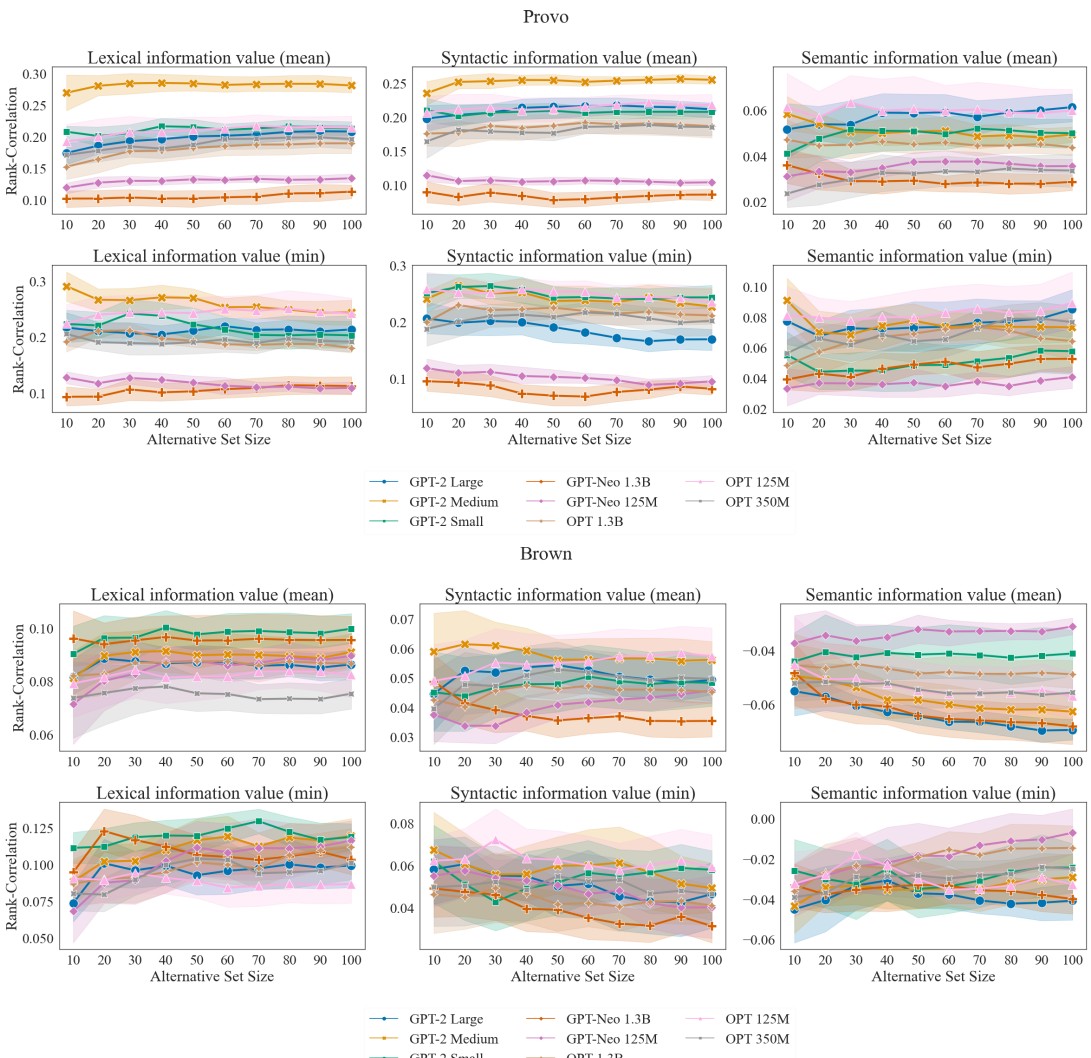

Figure 5: Spearman correlation between information value and average reading times (length-normalised).

correlation $\rho$ obtained between the information values for each parameter setting; strong pairwise correlation indicates that information value is robust to the varying parameter. Results are displayed in Figures 6 and 7.

Information value defined as lexical, syntactic, and semantic distance becomes highly robust as alternative set size increases; mean correlations between decoding strategies for each model converge towards perfect correlation as alternative set size increases. This pattern holds for all datasets. Decoding strategies do not produce much variation across correlations, regardless of alternative set size (see confidence intervals in Figures 6 and 7). For mean-based definitions and models, information values generated from different decoding strategies correlate at strengths $> 0.8$ from sets with fewer than 50 alternatives. Although their mean correlations still converge to 0.9, the dialogue datasets are slight exceptions.

As expected, correlations between parameter settings for min-based distances are more variable. Although they converge to weaker correlations as alternative set size increases when compared to mean-based distances, we still find strong to very strong correlations between decoding strategies for large alternative sets across all models.

## E  Selecting the Best Information Value Estimators

For each corpus and each surprisal type (lexical, syntactic, semantic), we select the estimator that yields the best Spearman correlation with the psychometric data. An estimator is a combination of model, sampling algorithm, and alternative set size. Psychometric data are in-context acceptability judgements for DailyDialog, Switchboard and Clasp, and the mean of all word reading times in a sentence for Brown and Provo. Table 5 shows the best estimators.

## F  Linear Mixed Effect Models

In this section, we include further details about the linear mixed effect models used in Sections 6 and 7. All results for single information value predictors are in Table 6. Results for the comparison with surprisal and joint models are in Table 2.

**Response variables.**  For PROVO, we use the total dwell time, i.e., the cumulative duration across all fixations on a given word. We filter away any observation that contains 'outlier' words, i.e., words

with a $z$-score $> 3$ when the distribution of reading times is modelled as log-linear (following Meister et al., 2021).

**Control predictors.**  Following Wilcox et al. (2020), we evaluate each model relative to a baseline model which includes only control variables. Control variables are selected building on previous work (Meister et al., 2021): we include solely an intercept term as a baseline for acceptability judgements and the number of fixated words for reading times. Meister et al. (2021) report similar trends when including summed unigram log probability or sentence length as baseline predictors of acceptability judgements, and word character lengths or word unigram log probabilities for reading times. For reading times, we also test sentence length as a predictor but baseline models that include, instead, the number of fixated words (readers sometimes skip words while reading) achieve higher log-likelihood.

## G  More Derived Measures of Information Value

We also tested the following measures derived from information value but found them to be less predictive than those in the main paper.

**Expected information value.**  The expected distance of plausible productions given a context $x$ from the alternative set:

$$\mathbb{E}(I(Y|X=x)) := \mathbb{E}_{a \in A'_x}\left[I(Y=a, X=x)\right] \tag{10}$$

We assume a uniform probability distribution over alternatives. This quantifies the uncertainty over next utterances determined by the context alone. Because the alternative set $A_x$ *is* the set of plausible productions given $x$, in practice, we compute expected information value using only one alternative set—both in the expectation $\mathbb{E}_{a \in A_x}$ and in the distance calculation $d(y, A_x)$.

**Deviation from the expected information value.** The absolute difference between the information value for the next utterance $y$ and the expected information value for any next utterance:

$$D(Y=y|X=x) := |I(Y=y|X=x) \\ - \mathbb{E}(I(Y|X=x))| \tag{11}$$

This quantifies the information value of an utterance *relative to* the information value expected for

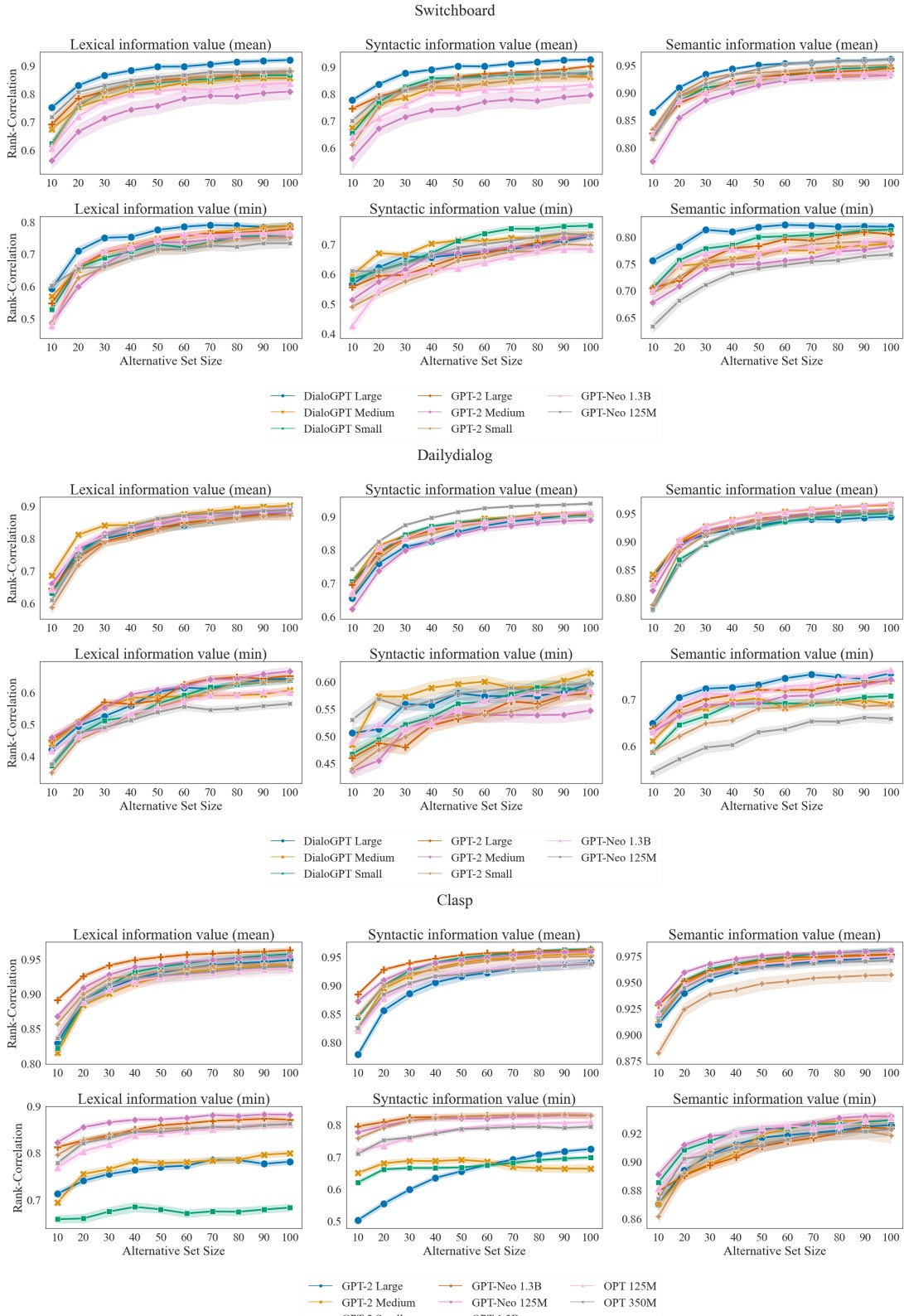

Figure 6: Intrinsic robustness evaluation on acceptability judgements corpora.

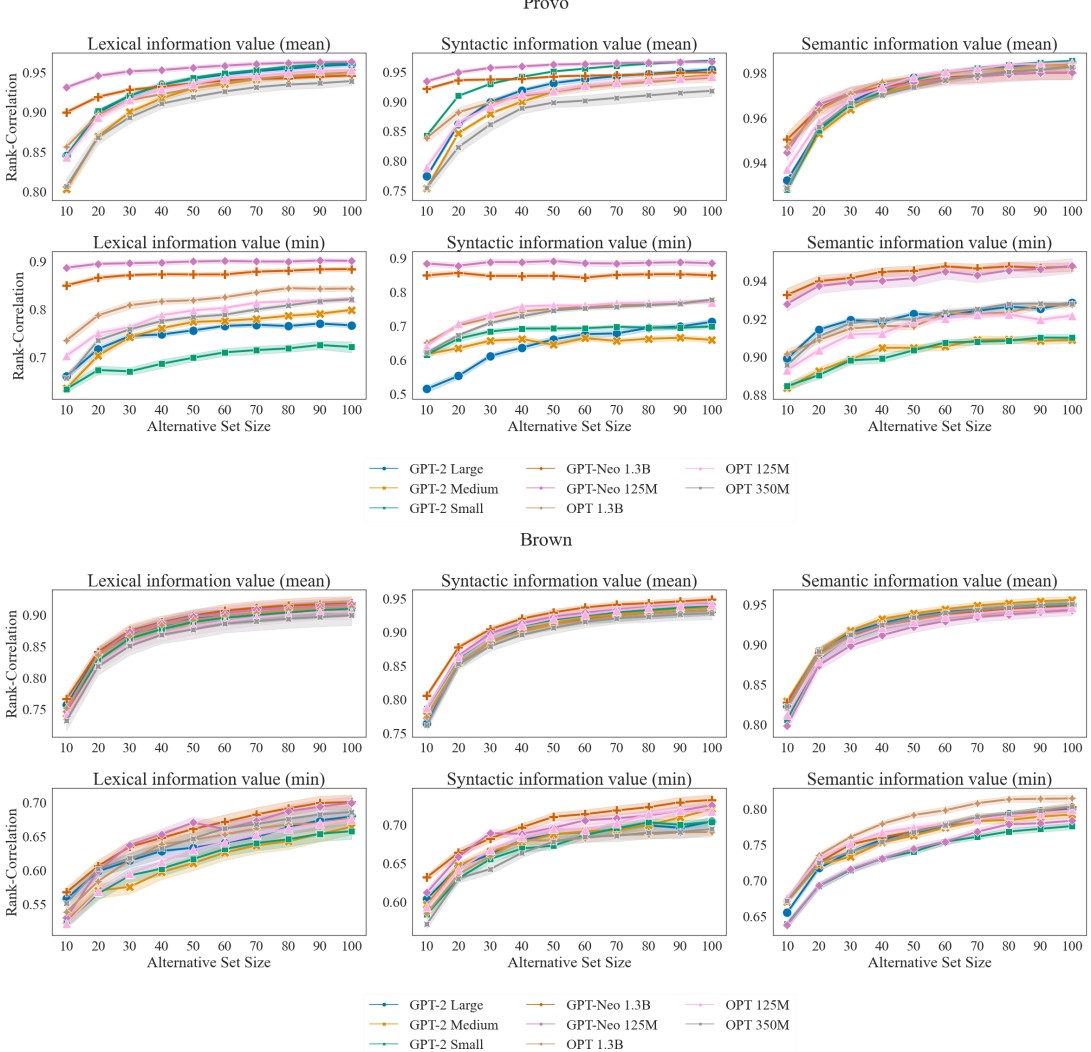

Figure 7: Intrinsic robustness evaluation on reading times corpora.

| Corpus | Level | Metric | Summary | $N$ | Language Model | Sampling | $\rho$ |
|---|---|---|---|---|---|---|---|
| | Lexical | Bigram | Min | 70 | DialoGPT Medium | Temperature 1.25 | -0.436* |
| SWITCHBOARD | Syntactic | POS bigram | Min | 100 | DialoGPT Small | Ancestral | -0.440* |
| | Semantic | Cosine | Min | 100 | DialoGPT Large | Temperature 1.25 | **-0.702*** |
| | Lexical | Unigram | Min | 80 | DialoGPT Small | Ancestral | -0.383* |
| DAILYDIALOG | Syntactic | POS trigram | Min | 90 | DialoGPT Large | Temperature 1.25 | -0.359* |
| | Semantic | Cosine | Min | 100 | GPT-2 Large | Nucleus 0.9 | **-0.584*** |
| | Lexical | Trigram | Min | 90 | GPT-2 Large | Temperature 1.25 | -0.210* |
| CLASP | Syntactic | POS Bigram | Min | 100 | GPT-2 Large | Nucleus 0.95 | **-0.234*** |
| | Semantic | Cosine | Min | 90 | OPT 1.3B | Temperature 0.75 | -0.221* |
| | Lexical | Unigram | Min | 10 | OPT 125M | Typical 0.3 | 0.379* |
| PROVO | Syntactic | POS Trigram | Min | 10 | GPT-2 Small | Nucleus 0.95 | **0.421*** |
| | Semantic | Euclidean | Min | 100 | OPT 125M | Nucleus 0.95 | 0.181 |
| | Lexical | Bigram | Min | 90 | GPT-2 Small | Typical 0.3 | **0.223*** |
| BROWN | Syntactic | POS Trigram | Mean | 10 | GPT-2 Medium | Typical 0.3 | 0.185* |
| | Semantic | Cosine | Min | 100 | GPT-Neo 125M | Nucleus 0.95 | 0.048 |

Table 5: Best information value estimator per corpus and metric. Spearman rank-correlation coefficients $\rho$, statistical significance ($p < 0.001$) is marked with a star. The highest correlations per dataset are in **bold**; the estimators (a combination of set size $N$, model, and sampling strategy) that generate them are taken as the 'best estimators' for that corpus and are used in Sections 6 and 7.

| Summ. | Level | Metric | SWITCHBOARD | | DAILYDIALOG | | CLASP | | PROVO | | BROWN | |
|---|---|---|---|---|---|---|---|---|---|---|---|---|
| | | | $\beta$ | $\Delta$LogLik | $\beta$ | $\Delta$LogLik | $\beta$ | $\Delta$LogLik | $\beta$ | $\Delta$LogLik | $\beta$ | $\Delta$LogLik |
| Mean | Lexical | Unigram | -0.273 | 1.874 | -0.683 | 2.152 | 0.594 | 0.206 | 2.309 | 9.967 | 2.409 | 9.679 |
| | | Bigram | -1.35 | 4.687 | -2.761* | 7.179 | -1.573 | 2.717 | 1.976 | 10.971 | 1.622 | 9.076 |
| | | Trigram | -2.401 | **8.315** | -1.843 | 7.089 | -2.514 | **5.857** | 1.982 | **12.169** | 1.891 | 11.974 |
| | Syntactic | POS Unigram | 0.204 | 0.605 | 3.399** | **6.707** | 0.914 | 0.106 | 1.902 | 8.413 | 0.958 | 6.627 |
| | | POS Bigram | 0.398 | 1.366 | 1.835 | 3.147 | -0.648 | -0.073 | 3.813** | 13.861 | 1.331 | 7.291 |
| | | POS Trigram | -0.159 | **2.488** | 0.767 | 2.505 | -2.011 | 2.274 | 5.527** | **21.798** | 1.475 | 8.194 |
| | Semantic | Cosine | -8.664** | 29.034 | -6.988** | 21.207 | -1.235 | 0.030 | 0.237 | 6.661 | 0.714 | 6.633 |
| | | Euclidean | -8.665** | 29.263 | -7.11** | 21.994 | -1.535 | 0.617 | 0.221 | **6.864** | 0.766 | **6.833** |
| Min | Lexical | Unigram | -3.927** | 7.701 | -4.454** | 10.244 | -0.866 | 0.005 | 2.219 | 9.649 | 2.39 | 9.059 |
| | | Bigram | -1.017 | 1.629 | -4.614** | **10.876** | -1.937 | 1.639 | 1.882 | 9.490 | 2.121 | 8.426 |
| | | Trigram | -1.774 | 3.396 | -1.969 | 3.311 | -2.757* | 3.714 | 1.997 | 10.337 | 3.689** | **13.913** |
| | Syntactic | POS Unigram | -0.52 | 0.927 | 0.356 | 1.985 | -2.931* | 4.915 | 5.45** | 21.187 | 1.947 | **8.633** |
| | | POS Bigram | 1.052 | 0.901 | -2.933* | 5.067 | -5.356** | **13.539** | 4.494** | 16.292 | 1.404 | 6.854 |
| | | POS Trigram | 0.758 | 0.993 | -3.26* | 5.732 | -3.104* | 4.124 | 4.956** | 18.394 | 1.362 | 6.706 |
| | Semantic | Cosine | -9.888** | **34.204** | -9.01** | **30.408** | -1.982 | 0.979 | 0.548 | 6.476 | 0.661 | 6.164 |
| | | Euclidean | -7.696** | 23.375 | -8.901** | 29.868 | -2.501 | **2.094** | 0.507 | 6.567 | 0.699 | 6.020 |

Table 6: Results of single-predictor linear mixed effect models: fixed effect coefficients $\beta$ and $\Delta$LogLik. Statistical significance of fixed effects is marked with one ($p < 0.01$) or two stars ($p < 0.001$). Information value estimates are obtained according to Equation 1. For each corpus and each level (lexical, syntactic, and semantic), the best $\Delta$LogLik is marked in **bold**. These are the level-specific metrics used whenever we talk about 'best predictors' in the main paper.

plausible productions given $x$. An analogous notion is the deviation of surprisal from entropy. The token-level version of this forms the basis of the local typicality hypothesis (Meister et al., 2023).

**Expected context informativeness.** The *expected informativeness* of context $x$ is the reduction in information value contributed by $x$ with respect to any plausible continuation:

$$\mathbb{E}(C(Y{=}y; X{=}x)) := \mathbb{E}(I(Y{=}y)) \\ - \mathbb{E}(I(Y{=}y|X{=}x)) \tag{12}$$

This quantifies the extent to which a context restricts the space of plausible productions. An analogous notion is the expected pointwise mutual information between $X = x$ and $Y$, where the value of $X$ is fixed. Similarly to out-of-context information value, out-of-context expected information value $\mathbb{E}(I(Y{=}y))$ is computed with respect to the alternative set $A_\epsilon$.