# OpenReview forum: "Information Value: Measuring Utterance Predictability as Distance from Plausible Alternatives"
_EMNLP/2023/Conference — EMNLP 2023 Main_

### Official Review · Reviewer_RsrJ · 2023-08-03

**Soundness:** 4

**Excitement:**

4: Strong: This paper deepens the understanding of some phenomenon or lowers the barriers to an existing research direction.

**Missing References:**

- Smith & Levy (2011): https://escholarship.org/uc/item/69s3541f This is relevant to the motivation for information value as a kind of predictability which isn’t captured by token surprisals. In this paper they find that human-predictable words are influenced not only by corpus frequencies but also by semantic similarity. This seems relevant to semantic information value.

**Paper Topic And Main Contributions:**

This paper proposes “information value” as an alternative to surprises a measure of predictability as it relates to human acceptability judgments and reading times. Information value is based on the distribution of distances between a token y and a set of alternative tokens A_x in context x generated from a language model. Three different distance metrics are tried, as well as two ways of aggregating the distances into a single number (mean and min). Information value is found to predict acceptability ratings better than surprisal, and is about as good as surprisal at predicting reading times (and complementary to it). The authors also examine some derived measures such as non-contextual information value.

**Questions For The Authors:**

- Why is a uniform mean used to reduce d(y, A_x) to a number? Why not weighted by probability, which would seem most natural? Or distance to the most probable member of A_x, etc.
- How long are the alternative utterances?

**Reasons To Accept:**

- The paper proposes a new concept that can be used in models of language processing.
- The proposed measure is easy to calculate and understand.
- The results show that the proposed measure is effective (modulo concerns about multiple comparisons below).
- The paper is well-connected with the processing literature.

**Reasons To Reject:**

- The large number of free parameters (which distance metric, which aggregation method, mean or variance for surprisal) is a bit unsettling. I think a more comprehensive reporting of the results in the main text than is shown in Table 1 is in order.
- The framing with respect to surprisal theory should be adjusted. Currently the introduction presents information value as an alternative theory of *predictability*, but it seems that it’s really an alternative theory of processing difficulty as reflected in acceptability judgments and reading times, contrasting with surprisal rather than measuring the same thing in a different way. In some parts of the intro it seems that information value is being proposed as an alternative way to measure *surprisal* (eg. “Proper estimation of the surprisal of an utterance would require computing probabilities over a … event space”), but it is not a measure of surprisal: I don’t see any formal relationship between d and surprisal. Rather, it seems that d is a new, separate and complementary predictor of processing difficulty.

**Reproducibility:**

4: Could mostly reproduce the results, but there may be some variation because of sample variance or minor variations in their interpretation of the protocol or method.

**Reviewer Confidence:**

5: Positive that my evaluation is correct. I read the paper very carefully and I am very familiar with related work.

**Typos Grammar Style And Presentation Improvements:**

Line 186: Should be alternative sets A_x, not |A_x|
Line 109-113: “and serves as a foundation for quantitative principles of language production and comprehension such as ERC and UID” — Only if the linking function from surprisal to processing difficulty is superlinear.

---

> ### Author Rebuttal · Authors · 2023-08-26
>
> Thank you for your comments, questions, and suggestions!
>
> **Large number of free parameters.** We will consider moving Table 5 to the main paper and including a more exhaustive version thereof in the Appendix.
>
> **The framing with respect to surprisal theory.** We think of surprisal (or Shannon information content) as a measure of predictability, where “predictable” is used out of jargon for an event that is regarded as likely to happen. Furthermore, we see surprisal theory as asserting that the processing effort incurred by a linguistic signal is proportional to the predictability of that signal in context as captured by the specific measure of surprisal. What we are proposing is an alternative measure of predictability—not of surprisal (that would indeed not make much sense, as it would be as stating that information value is a measure _of a measure_ of predictability). We see information value as a new way to measure predictability, next to surprisal.
>
> **Why a uniform mean?** For simplicity, we chose to treat generators as sources of alternative sets and to disconnect information value from the language model probabilities. We agree it would also be interesting to study other reduction strategies, such as the weighted mean, the distance from the most probable alternative, or strategies that rely on the clustering of alternatives (e.g., Kuhn et al., 2023). We will include this point in the discussion.
>
> **How long are the alternative utterances?** The mean and standard deviation of distributions of length as number of words for all alternatives generated by the best estimator for each dataset (as described in lines 415-425) are as follows:
> - Switchboard: 13.90 +- 15.01 (note that this distribution is heavily skewed towards short utterances due to the high proportion of backchannel responses in conversational speech)
> - Dailydialog: 23.53 +- 4.16
> - Clasp: 18.72 +- 12.24
> - Provo: 18.87 +- 12.38
> - Brown: 17.99 +- 11.71
>
> We will include these statistics in the camera-ready, probably in the Appendix for reasons of space. In addition, we will release our large set of generated alternatives as we believe characterising these to be an interesting direction to explore.
>
> **Missing References and Presentation Improvements**
> We will include Smith & Levy (2011) and further specify the connection with the ERC and UID principles in the camera-ready.

---

### Official Review · Reviewer_i1qv · 2023-08-05

**Soundness:** 4

**Excitement:**

4: Strong: This paper deepens the understanding of some phenomenon or lowers the barriers to an existing research direction.

**Paper Topic And Main Contributions:**

This paper describes a new measure of information to replace Shannon information (surprisal).  Evaluations find it to be complementary to surprisal.

**Questions For The Authors:**

This has nothing to do with my evaluation, but I think your weighted distance measure might be equivalent to surprisal (log prob) of a multivariate Gaussian (exponential) probability model centered on the mean semantic/syntactic/lexical vector, where the distribution of distances is doing the job of Mahalanobis distance (distance in terms of standard deviations) in a Gaussian.  Is it?  If so, it might be an intuitive way to explain why this new information measure adds and subtracts like Shannon information.

**Reasons To Accept:**

This new, more interpretable measure of information may allow finer-grained exploration of the relationship between expectation and human reading time.

I did not notice any issues with the evaluation.

**Reasons To Reject:**

The predictor is only evaluated against utterance-level surprisal, but one advantage of surprisal over this new predictors is that it can be estimated at the token level and might be a better predictor when estimated that way, whereas I'm not sure this new predictor can.  If some of the information value measures can be estimated at the token level, are they as good?

**Reproducibility:**

4: Could mostly reproduce the results, but there may be some variation because of sample variance or minor variations in their interpretation of the protocol or method.

**Reviewer Confidence:**

4: Quite sure. I tried to check the important points carefully. It's unlikely, though conceivable, that I missed something that should affect my ratings.

---

> ### Author Rebuttal · Authors · 2023-08-25
>
> Thank you for your interesting comments!
>
> **Token-level comparison of information value vs. surprisal.** This paper focuses on utterance predictability (and utterance-level psychometric variables) but in principle, our measure of predictability can be extended to the token level. This can be done, for example, by measuring the semantic distance between contextualised word representations. We are very curious to see, in follow-up studies, how token-level information value would fare in comparison to token-level surprisal.
>
> **Surprisal of a multivariate Gaussian.** We need to think more about this connection, but thank you for the hint :)

---

### Official Review · Reviewer_TJwo · 2023-08-09

**Soundness:** 4

**Excitement:**

3: Ambivalent: It has merits (e.g., it reports state-of-the-art results, the idea is nice), but there are key weaknesses (e.g., it describes incremental work), and it can significantly benefit from another round of revision. However, I won't object to accepting it if my co-reviewers champion it.

**Missing References:**

L106 seems to misrepresent Hale (2001) as a work that uses Markov chains for estimation of surprisal; he uses an Earley parser.

**Paper Topic And Main Contributions:**

This paper introduces the notion of information value, which is a measure for the predictability of a sentence. It is distinguished from other measures based on surprisal in that it accounts for alternative plausible sentences: information value is a distribution over the distance between the incumbent sentence and a set of plausible alternative sentences given the context, and precise measures are obtained as summary statistics of this distribution. They perform a grid search type study to find how information value correlates with and predicts acceptability judgements and reading times. Their results indicate that information value provides a strong predictor of acceptability judgements, particularly when using a semantic distance metric. Overall I think this is an interesting experimental study that has some room for improvements when it comes to presentation.

**Questions For The Authors:**

A Is there a motivation behind the terminological choice “information value”? This term would suggest a close relation to information theory, but as far as I can tell there is no such relation described in the paper.

**Reasons To Accept:**

The experimental setup is sound and has a high degree of breadth, experimenting with several language models, distance metrics, alternative set sizes and sampling techniques. Measuring predictability by making use of alternatives is an interesting idea, appears novel and provides insight into what type of information is considered under different aspects of human language processing.

**Reasons To Reject:**

~~I have some concerns regarding the information value metric as presented in Sec 3. In particular, it is unclear to me how information value is a probability distribution. If it is, I assume you would mean to normalize your distance metrics such that your function fulfills the axioms of a probability distribution. Furthermore, it seems to me that the number of alternatives in your alternative set and how these are sampled would have a significant effect on your distribution, which I miss a discussion on. I assume that the event space of your distribution is the sentences in the alternative set but the interpretation is not entirely clear to me. Overall, I find this key section of the paper hard to read and believe it would benefit from another round of revision.~~

Furthermore, I would have appreciated a more careful discussion on how this type of approach solves some of the problems with previous methods. In the introduction you write that aggregating token-level surprisal estimates leads to different realizations of the same concept or communicative intent competing for probability mass, but why is this a problem? And how does your method solve it? As far as I can tell there is no technique used here that in some way guarantees that the different utterances in the alternative set would be distinct in terms of communicate intent.

**Reproducibility:**

4: Could mostly reproduce the results, but there may be some variation because of sample variance or minor variations in their interpretation of the protocol or method.

**Reviewer Confidence:**

2: Willing to defend my evaluation, but it is fairly likely that I missed some details, didn't understand some central points, or can't be sure about the novelty of the work.

**Typos Grammar Style And Presentation Improvements:**

Sec 5.1 discusses correlations of the information values with psychometric variables compared to those of previous work using surprisal. It would be useful to have a table displaying the results of previous works as well as your own experiments using surprisal with more details than Table 1. The reading times discussion further lacks any references to previous work that the reader can refer to for comparison.

L196 you may want to refer to a specific paragraph
L206-8 is confusing to me. What low dimensional properties?
L229-30 it would be useful to see an explanation on what is meant by “under a uniform distribution of alternatives”.
L232-4, is that a sneak peek of the results?
L248 I would change notation since Y is not related to the target utterance y.

---

> ### Author Rebuttal · Authors · 2023-08-26
>
> Thank you for your comments and questions. We try to address them below point by point.
>
> > I have some concerns regarding the information value metric as presented in Sec 3. In particular, it is unclear to me how information value is a probability distribution. If it is, I assume you would mean to normalize your distance metrics such that your function fulfills the axioms of a probability distribution.
>
> Given a context $x$, an utterance $y$, a probability distribution $p_{Y|X=x}$ prescribed by a language model, and a choice of distance metric $d$, $d(y, Y)$ is a real random variable that captures the distribution of distances between $y$ and any output drawn conditionally from $p_{Y|X=x}$. We propose sampling a fixed number of alternatives, which make up the alternative set $A_x$, and define information value as the distribution of distances between $y$ and all the alternatives in $A_x$ (Eq. 1). The word "distribution" here means… a distribution, i.e., “simply a collection of data, or scores, on a variable” (Urdan, 2016); in our case the variable is distance (cosine distance, n-gram overlap, or POS n-gram overlap). In other words, information value is not a probability distribution (with pdf, cdf, etc.) and indeed this is never stated in the paper.
>
> > Furthermore, it seems to me that the number of alternatives in your alternative set and how these are sampled would have a significant effect on your distribution, which I miss a discussion on.
>
> This is certainly true. As such, we have a subsection (Section 5.2) as well as two appendices (B and E) that investigate the sensitivity/robustness of our information value estimates to estimator parameters (i.e., the number of alternatives, the language model, and the sampling strategy).
>
> > Overall, I find this key section of the paper hard to read and believe it would benefit from another round of revision.
>
> Thank you for the feedback, we will try to clarify all the aspects above using the additional space available in the camera-ready.
>
>
> > Furthermore, I would have appreciated a more careful discussion on how this type of approach solves some of the problems with previous methods. In the introduction you write that aggregating token-level surprisal estimates leads to different realizations of the same concept or communicative intent competing for probability mass, but why is this a problem? And how does your method solve it? As far as I can tell there is no technique used here that in some way guarantees that the different utterances in the alternative set would be distinct in terms of communicative intent.
>
> In the Introduction, we mention two downsides of token-level aggregates: (1) different realisations compete for probability mass, (2) different dimensions of predictability are conflated. We acknowledge that we give more weight to our solution to the second issue throughout the paper (with information value allowing us to disentangle different dimensions of predictability thanks to the use of interpretable distance metrics). Regarding the first issue, consider the following case. Let $x$ be a dialogue context and $y$ the true upcoming utterance “I gotta go”. Say the language model assigns a probability p = 1e-7 to $y$ (e.g., because the use of “gotta” is not in line with the formality level of the dialogue so far) and that it assigns the following probabilities to alternative continuations: p(“I have to go”) = 0.001; p(“I must go”) = 0.0005; p(“I need to go”) = 0.0005. The LM’s surprisal for “I gotta go” will arguably be too high in this scenario. Now consider using information value for the same case. Utterances like “I have to go”, “I must go”, and “I need to go” will be much more likely to be sampled in the alternative set and, in turn, the semantic predictability of “I gotta go” more likely to not be underestimated (its lexical predictability may still be rather low, which again attests to how information value addresses problem 2). Now, due to the stochasticity of the sampling algorithms, to the limited size—in practice—of alternative sets, and to the fact that the sampling algorithms we test are themselves token-level, it might still be possible that none of the three higher likelihood alternatives might be sampled because of their “competition” for probability mass. Here, utterance-level decoding algorithms (e.g., Eikema and Aziz, 2022; as cited in the paper) might be of help. Furthermore, we do not aim to guarantee that alternatives are distinct in terms of communicative intent as certain intents are simply more likely for particular contexts, but rather we aim to account for variation at this level of abstraction more explicitly than is currently done by aggregates of token-level surprisal estimates. There exist some very recent approaches for monitoring the variety of intents in the LM’s output space (e.g, Kuhn et al. 2023; Stasaski and Hearst 2023; as cited in the paper); we think they are a promising solution to this problem. In any case, *in the camera-ready, we will tone down our claims about information value solving problem 1 and discuss them more carefully.*
>
> > Is there a motivation behind the terminological choice “information value”? This term would suggest a close relation to information theory, but as far as I can tell there is no such relation described in the paper.
>
> We chose “information value” to signal the connection of our measure of predictability to the information-theoretic notion of information content or surprisal, while avoiding the terms “information content” and “surprisal”. The relation between predictability and information (Hale, 2001; Genzel and Charniak, 2002; Jaeger and Levy, 2007, as cited in the paper) is described in the first paragraph of the Introduction (lines 27-37). Moreover, the term “information value” resonates with attempts to develop a quantitative measure of information that go beyond the probability of outcomes (Howard, 1966). We also considered referring to our measure of predictability as “surprise”, resonating, this time, with the vocabulary of active inference and the free variational energy principle (Friston, 2010; Friston et al., 2017).
>
> > Presentation Improvements
>
> Finally, we appreciate your feedback regarding presentation improvements. We provide references to a number of previous results in reading time predictions in the Reading Times section (4.2.2) but will aim to present these more clearly in a camera-ready version, as well as clarify the other points of definition that you highlighed (“low-dimensional properties”, and “uniform distribution”).
>
> ———
> Urdan, T.C., 2016. _Statistics in plain English_. Routledge.
>
> Howard, R.A., 1966. Information value theory. _IEEE Transactions on systems science and cybernetics_, 2(1), pp.22-26.
>
> Friston, K., 2010. The free-energy principle: a unified brain theory? _Nature Reviews Neuroscience_, 11(2), pp.127-138.
>
> Friston, K., FitzGerald, T., Rigoli, F., Schwartenbeck, P. and Pezzulo, G., 2017. Active inference: a process theory. _Neural computation_, 29(1), pp.1-49.

---

### Official Review · Reviewer_fDKN · 2023-08-10

**Soundness:** 4

**Excitement:**

3: Ambivalent: It has merits (e.g., it reports state-of-the-art results, the idea is nice), but there are key weaknesses (e.g., it describes incremental work), and it can significantly benefit from another round of revision. However, I won't object to accepting it if my co-reviewers champion it.

**Paper Topic And Main Contributions:**

This paper tests the predictability of utterances, which could be considered as an incremental work toward ERC/UID theory.
One of the merits of this work is that, it puts forward a new method to measure the predictability of utterance(or beyond it), which is based on the distance among the "ground truth" next utterance and the "candidate sentences" generated by LLM(the distance measurement is cosine similarity score according to my reading).
The effectiveness of the new method is further tested by two types of experiments: acceptability measurement and reading time measurement based on eye gaze tracking.
Based on the experiments, this paper further proves the correlation between Information Value(represented as distance in three different linguistic aspect, namely syntactic, lexical and semantic) and acceptability/predictability in an utterance level, as well as a more advanced level, i.e., dialogue act level.
Overall, this paper introduces a new methodology, in which its potential usage on the evaluation of NLG could be expected.
It orients itself quite well in the direction of computationally-aided linguistic analysis.

**Questions For The Authors:**

In figure 1 caption, it says that 11 sampling strategies are used while in the main text, only 6 sampling methods are mentioned, it is necessary that author(s) give an explanation on that.

**Reasons To Accept:**

1.Solid and complete overview of previous relevant literature is highly appreciated in term of long paper.

2.The experiments performed in this paper is thorough and have taken relevant previous works into full consideration.

3.Although the usage of distance is not a new idea, the way of using it to represent information value instead of using entropy value is interesting and according to the experiment results, quite promising.

4.The predictability of utterance is evaluated from three perspectives: lexical level, syntactic level and semantic level, which is linguistically thorough and coherent.

Overall, this is an information-intense while interesting paper and I found no strong reason to reject it, it is more about either having it in finding or main venue.

**Reasons To Reject:**

1. Have the author(s) thought about the reason why, information value is "stronger predictor" for dialogue(Complementarity in page 7 or discussion in page 8), is there any already existing linguistic theory which could explain it. If so, adding that will make this one a stronger paper.

2.It turns out to me that different information value serves as a strong predictor among the 5 chosen corpora, for example, in PROVO it is the syntactic information value. Again have the author(s) already had a potential hypothesis for this phenomenon?Is it practical to do a linguistic analysis on this 5 corpora to find the reason?

3. Is it possible that by increasing the set size of A_(x),  the generated sentences sampled from "ancestral sampling" could have already covered some/all of the samples from other sampling methods, e.g., "temperature sampling"? As far as I know, both of the two mentioned  sampling methods are based on the conditional probability, while typical sampling is a comparatively new sampling method which could cut off the dependence on the conditional probability to some degree and helps to generate more creative and diversify next sentences instead of entering receptive loop(Meister 2023).
Based on that,I would suggest the author(s) making a statistic report about the distributions of generated sentences from different sampling methods and maybe then making a selection of just two representative sampling methods based on the observation. Also a reference to temperature sampling is needed.

**Reproducibility:**

4: Could mostly reproduce the results, but there may be some variation because of sample variance or minor variations in their interpretation of the protocol or method.

**Reviewer Confidence:**

4: Quite sure. I tried to check the important points carefully. It's unlikely, though conceivable, that I missed something that should affect my ratings.

**Typos Grammar Style And Presentation Improvements:**

Below are the two suggestions related to presentation improvements:


1.The abstract should be definitely improved so that the readers could grasp the skeleton of the story of the paper better and more easily, especially in term of such long paper.

2.One of the important concerns towards the selected corpora is that, they are all English Corpus. It is necessary to mention this point in the limitation section, i.e.,  about the potential consequence of using only English language data.

---

> ### Author Rebuttal · Authors · 2023-08-26
>
> Thank you for your review! We answer your questions below.
>
> 1. We think information value is a stronger predictor of dialogue acceptability judgements because, when estimated via semantic distance metrics, (1) it can more robustly identify semantically incoherent dialogue utterances and (2) it can capture expectations over linguistic signals at higher levels of abstraction, such as speech act types (Austin, 1962; Searle 1969, 1975), more explicitly than token-level surprisal aggregates. We have mentioned these two reasons in lines 496-500 and 596-602, but we agree they can be given more prominence! We will discuss them more explicitly in the camera-ready, including the citations to speech act theory.
> 2. We present our hypotheses about the predictive power of different linguistic levels in the introductory paragraphs of Sections 6.2 (lines 487-504) and 6.3 (lines 540-552). These are based on linguistic and psycholinguistic theory regarding the potentially different processing requirements of the different tasks studied in this work, but also on the construction of the corresponding datasets (e.g.lines 500-504). They are largely confirmed by our analysis.
> 3. Except for ancestral (unbiased) sampling, all other sampling algorithms deployed (temperature sampling, nucleus sampling, and typical sampling) bias away from the underlying conditional distribution of the language model and make the resulting probability mass function hard, or impossible, to characterise. This is why we do not see any principled reason to restrict the set of sampling algorithms. *In practice*, we do observe some variability among sampling strategies (see confidence intervals in Figure 1) yet we find that this variability has no impact on predictive power (see again Figure 1), so we select only one sampling strategy per corpus for our analyses in Sections 6 and 7 (lines 417-421).
> Beyond the confines of this paper, we agree that continued analysis of the differences between sampling strategies is important. We have presented our preliminary findingsin Appendix E (intrinsic evaluation) where we evaluate the variability of alternative set sizes for each sampling strategy. We think your suggestion of making a statistical report about the distributions of sentences generated via different sampling algorithms is an interesting piece of follow-on work. We will publicly release our large dataset of generations to encourage this line of research.
>
> Regarding your question about the caption of Figure 1: we use 4 sampling methods with varying parameter configurations: 1 configuration for ancestral sampling (no parameters here), 2 for temperature sampling, 4 for nucleus sampling, and 4 for typical sampling—for a total of 11 “sampling strategies”.
>
> Lastly, thank you also for your presentation improvement suggestions. We will consider extending the abstract in the camera-ready, and we will include the English-only limitation in the respective section.
>
> ———
> Austin, J.L., 1962. _How to do Things with Words_, 2nd edition, J.O. Urmson and M. Sbisá (eds.), Cambridge, MA: Harvard University Press.
>
> Searle, J., 1969. _Speech acts: an essay in the philosophy of language_. Cambridge: Cambridge University Press.
>
> Searle, J., 1975. A taxonomy of illocutionary acts. In K. Gunderson (ed.), _Language, Mind and Knowledge_, Minneapolis, MN: University of Minnesota Press, 344–369.

---

### Meta-Review · Area_Chair_oecy · 2023-09-19

**Recommendation:** 5

**Metareview:**

This paper combines fundamental linguistic and cognitive questions with sound modeling. It is well situated in previous literature and makes insightful and sound contributions.

---

### Decision · Program_Chairs · 2023-10-07

**Decision:**

Accept-Main

**Comment:**

This paper combines fundamental linguistic and cognitive questions with sound modeling. It is well situated in previous literature and makes insightful and sound contributions.